# Why big brains? A comparison of models for both primate and carnivore brain size evolution

Helen Rebecca Chambers[ID]¹*, Sandra Andrea Heldstab², Sean J. O'Hara¹

1 School of Science, Engineering & Environment, University of Salford, Salford, Greater Manchester, United Kingdom, 2 Department of Anthropology, University of Zurich, Zurich, Switzerland

* h.r.chambers@edu.salford.ac.uk

**Data Availability Statement:** All relevant data are within the manuscript and its Supporting Information files.

**Funding:** The author(s) received no specific funding for this work.

## Abstract

Despite decades of research, much uncertainty remains regarding the selection pressures responsible for brain size variation. Whilst the influential social brain hypothesis once garnered extensive support, more recent studies have failed to find support for a link between brain size and sociality. Instead, it appears there is now substantial evidence suggesting ecology better predicts brain size in both primates and carnivores. Here, different models of brain evolution were tested, and the relative importance of social, ecological, and life-history traits were assessed on both overall encephalisation and specific brain regions. In primates, evidence is found for consistent associations between brain size and ecological factors, particularly diet; however, evidence was also found advocating sociality as a selection pressure driving brain size. In carnivores, evidence suggests ecological variables, most notably home range size, are influencing brain size; whereas, no support is found for the social brain hypothesis, perhaps reflecting the fact sociality appears to be limited to a select few taxa. Life-history associations reveal complex selection mechanisms to be counterbalancing the costs associated with expensive brain tissue through extended developmental periods, reduced fertility, and extended maximum lifespan. Future studies should give careful consideration of the methods chosen for measuring brain size, investigate both whole brain and specific brain regions where possible, and look to integrate multiple variables, thus fully capturing all of the potential factors influencing brain size.

## Introduction

Brain size varies considerably amongst mammals; substantial variation is seen among primates, where brain size varies almost a thousand-fold across the order [1]. The adaptive value of such variation has come under extensive scrutiny over the past few decades and yet despite considerable research effort, much uncertainty remains regarding the selection pressures responsible.

Frequently proposed to explain variation in brain size are factors related to the physical environment, such as diet and home range size, as well as factors related to the social

**Competing interests:** The authors have declared that no competing interests exist.

environment, such as group size and pair-bondedness. Ecological hypotheses mainly involve investigating the cognitive demands associated with foraging [2–7], as foraging is considered mentally demanding due to the pressure of managing, processing and remembering spatial and temporal information about resource availability [8–12]. Additionally, differing home range size is of interest to researchers due to the supposed cognitive demands imposed by larger home ranges, such as processing requirements of navigating spatially-complex information, especially in terms of food availability, location and distribution [9, 13–15]. This has resulted in many studies investigating the cumulative effects of the physical environment on encephalisation, with a specific interest in diet [16–20], home range [13, 14], foraging techniques [12, 21–23] and behavioural responses in a fluctuating environment [24].

In contrast to ecological hypotheses, the social brain hypothesis (SBH) suggests sociality – specifically the cognitive demands of tracking, negotiating and maintaining social relationships – to be the main driving force behind variation in primate brain sizes [25–27]. The study of primates lends credence to this hypothesis, with brain size found to correlate with many social proxies, such as social group size [28], tactical deception [29] and grooming clique size [30]. Evidence has since not been limited to studies of the primate lineage, with corroboration coming from research on spotted hyenas [31, 32] as well as other carnivorans [33–35], ungulates [36, 37], birds [38–40], and some fish species [41–43]. The focal point of much of the early work investigating sociality was social group size, due to the information-processing demands group of increasing sizes are thought to incur [26]. However, the use of this proxy for measuring social complexity has been criticised [44] and instead, focus has shifted to the consequences of varying levels of relationship complexity [45], and toward investigating the influence of pair-bondedness [27, 46–48]. This developed from the proposition that relationship quality [45, 49] connotes cognitive complexity.

Despite the hypothesis receiving considerable support in the past, more recent investigations have failed to find statistical support for a link between brain size and sociality [14, 19, 20, 50, 51]. Instead, it appears there is now substantial, strong, phylogenetically-corrected comparative data reinforcing the assertion that diet better predicts brain size in both primates and carnivores [14, 20, 52]. In addition, the obvious exceptions to the SBH, taxa that possess large brains but that are not considered social, suggest factors other than sociality may be influencing brain size [19, 53, 54]. For example, if sociality is to be accepted as the causal agent for increased encephalisation in mammals, it should be widespread across bears and musteloids, that show similar encephalisation increases to Canidae [55].

A further problem to have dogged comparative analyses of brain evolution is deciding on the correct brain measure. Whilst most studies tend to focus on whole brain size, even this can become an arduous task since there is little clarity in the literature regarding the most appropriate body size correction factor, making decisions on the correct method of choice challenging. Typically, cognitive abilities are estimated using relative brain size, by taking residuals from a regression curve or calculating encephalisation quotients [56, 57]. This became the method of choice when brain and body size were found to be tightly coupled allometrically across vertebrates; therefore, accounting for this allometric relationship became of great importance [35, 58]. However, the use of relative brain size and encephalisation quotients is not without criticism; for example, using residuals as data points in regression models has been discouraged, as the estimates produced are thought to be biased, which influences subsequent analyses [59, 60]. Encephalisation quotients possibly reflect the result of recent decreases or increases in body size [61], evidence for such was uncovered by Swanson et al. [19]. They found carnivore brain size to lag behind body size over evolutionary time, therefore hinting that the use of brain estimates may be a poor representation of carnivore brain size. However, no evidence for a lag is found for primates [62], suggesting a taxonomic difference for this

group. Alongside this, the prevalent use of relative brain size is thought to possibly hide other evolutionary pathways which may be influencing adaptations in body mass [63]. For example, a recent analysis of mammalian brain size found the brain-to-body relationship to uncover more than just selection on brain size, indicating relative brain size measures, both residuals and EQ scores, are not accurately capturing brain size variation, and are not suitable for comparisons across species with different evolutionary histories [64]. Thus, van Schaik et al., [65] suggest the use of encephalisation quotients should be avoided in future studies, as EQs repeatedly fail to accurately predict brain size, and thus, varying levels of cognitive ability. For example, Deaner et al., [57] found absolute brain size measures, over statistically produced methods i.e., residuals, to be the best predictors of primate cognitive abilities.

Alongside the use of total brain size, particular emphasis has been put on specific brain regions in recent years. The social brain hypothesis suggests the neocortex is the brain structure of interest, with primates' large brains thought to be mainly the consequence of a dramatic increase in neocortical volume [66–68]. The neocortex is thought foremost responsible for the processing of more demanding cognitive and social skills [69, 70] associated with intelligent and flexible behaviour [61]. Neocortical enlargement in primates is thought to be partly due to selection on visual mechanisms [71] which is important for frugivorous species, for example when needing to distinguish between fruits of different colours [72–74] or when manipulating small fruit and seeds that require fine motor coordination [75]. Alternatively, these visual mechanisms are thought to be important for processing complex and rapid social interactions, including understanding facial expressions, gaze direction and posture [76], suggesting that neocortical modifications associated with complex social lives primarily involve areas specialised for visual processing of social information [77]. In primates, the neocortex constitutes a substantial portion of the brain [66, 67] and a large proportion of the neocortex is comprised of visual information processing areas [71, 78, 79], which is thought to explain links found between frugivory and brain size (see [20]), as well as social group size and neocortex volume (see [1, 71]).

Alongside research into the neocortex, attention is focused on the cerebellum and its importance. The cerebellum was found to co-evolve with the neocortex [61], with a significant correlation found between these two brain regions [80]. Increased cerebellar volume is suggested to allow increased processing capacity, in terms of enhanced motor abilities and manipulative abilities [81, 82]. For example, in primates positive correlations are found between cerebellum volume and extractive foraging techniques [1], as well as the presence of neural activation in the cerebellum during tool use in monkeys [83].This highlights the influential role played by the cerebellum in technical intelligence [84]. Alongside this, the cerebellum is thought to be important in social intelligence [1], particularly in terms of the links between sensory-motor control and social interactions and understanding [85, 86]. Indeed, it is now thought the expansion of the cortico-cerebellar system is the primary driver of brain expansion in anthropoid primates [87], suggesting the increased behavioural complexity in mammals could be partly explained by selection on the cerebellum [88]. So much so, that Fernandes et al., [89] found residual cerebellar size to be the most appropriate proxy when compared to a measure of general intelligence; as cerebellar models produced the most similar model fit results when compared to those produced using a measure of general intelligence.

Here, using data aggregated from the literature the relative importance of social, ecological and life history traits are assessed on both overall encephalisation and specific brain regions, and different models of brain size evolution are tested. Considerable attention has been paid to primate brain evolution (e.g., [14, 20, 90, 91]), perhaps the result of the anthropocentricism and since there are substantial data available on this taxonomic group making comparative tests easy to implement. Likewise, carnivorans are also now receiving attention (e.g., [19, 88,

92, 93]) since variation in their brain and body size, and ranging social and physical environments, makes them excellent models for these tests too. Indeed, most of the literature surrounding brain size hypotheses is based on analyses of these two groups.

One aim here, therefore, is to provide greater clarity within these two groups. Integrating predictors into a framework which allow the assessment of multiple hypotheses simultaneously has become increasingly important for tests of brain evolution [94, 95]. Therefore, phylogenetically-corrected generalised least squares (PGLS) models are used here to account for shared evolutionary history, whilst assessing the potential variables influencing encephalisation. We use a recently updated phylogenetic tree to ensure phylogenetic relationships are contemporary. Further, the inclusion of multiple variables allows the comparison of multiple hypotheses, as well as models of varying complexity. While brain data are available for more taxa than are included in our dataset, we found some limitations on the completeness of the necessary covariate data. We present here our analyses of two orders where complete datasets with all covariates are available for all species, ensuring the most robust model comparisons.

## Methods

### Data collection

**Brain data.** Endocranial volume (ECV) and body mass data for primates (n = 83) and carnivores (n = 85) were compiled from multiple sources (see supplementary material). Volumes were matched for species composition and predictor variables, and whilst this resulted in smaller sample sizes when compared to available brain data, in doing so it provided a complete dataset with all covariates available for all species, better enabling robust analyses. ECV data were preferred over brain mass data since it is thought ECV provides a more reliable estimate of brain size, due to the influence of preservation techniques on brain mass [96]. The standard technique for estimation of ECV is through filling the cranium with beads (or similar), which is then measured using a graduated cylinder or by weighing the beads and converting the weight to volume [96]. Neocortex and cerebellum volumes were also collated, where available, for both primates (Neo = 52, Cere = 49) and carnivores (Neo = 44, Cere = 38). Regional brain volumes are commonly measured using one of two different techniques: virtual endocasts (e.g., [19]) or physical sectioning of the individual brain volumes using paraffin and staining substances (e.g., [97]). When sourcing whole and regional brain volumes these measurement methods were considered to ensure the data were comparable; for example, all ECV data sources used common measurement techniques (as described above) making the whole brain data comparable across multiple studies.

**Social data.** Both social group size and social cohesion data were collected for primates and carnivores. Group size–based on the simple principle that as group size increases the information-processing demands [26] and corresponding internal structures [98, 99] should also increase – became perhaps the most commonly used proxy for social complexity. Despite this, the use of this proxy has been criticised as it is often considered crude, weak, and not always relevant [44]. Greater attention is now paid to differing levels of relationship complexity [45] often indicated through the presence of pair-bonds [27, 34, 100]. Therefore, to ensure the influence of sociality was fully captured, alongside group size, a social cohesion proxy was used: a categorisation system ranging from 1) being primarily solitary living aside from breeding seasons, 2) pair-living, 3) fission-fusion societies, to 4) being obligatorily social (e.g., [91, 101]). This index aims to better encapsulate sociality, rather than relying solely on group size numbers.

**Ecological data.** Four ecological variables were chosen for analysis: dietary categories, dietary breadth, habitat variability and home range size. Dietary categories were assigned

following previous designations in the published literature (see supplementary material for sources) and included six different categories: carnivorous, herbivorous, piscivorous, folivorous, frugivorous and omnivorous. Alongside this traditional classification system, we also used dietary breadth, estimated using the total number of food sources used by a species, with data taken from [102]. This included a total of 10 different food types: invertebrates, mammals and birds, reptiles, fish, unknown vertebrates, scavenge, fruit, nectar, seed or other plant material, marked either as absent (0) or present (1). For this dataset, this resulted in a dietary breadth scale of one to six. Habitat variability, another ecological measure, was formed using data from the IUCN Red List [103], based on the total number of habitat-types used by a species, following the same habitat classification system used in the IUCN Red List. Additionally, home range size data were collected. By including variables related both to diet and habitat, it allowed greater incorporation of possible variables within the physical environment affecting brain size. We acknowledge, however, such proxies measure ecological variability in the broadest sense, often producing large margins of error. Notwithstanding, these measures are widely used, due to data availability and since data consistency across groups can be achieved.

**Life-history data.**   Life-history variables have been found to be critical in counterbalancing the costs of increased brain size and facilitating the growth of large brains [104]. In fact, they appear to be influencing the potential adaptive pathways available to a species [94], for example in terms of balancing shifting developmental and maturation periods. Developmental costs are also thought to influence correlations between specific primate brain structures and life history variables, with the neocortex most strongly correlated with gestation length, and the cerebellum with juvenile period length, suggesting that these brain regions exhibit distinct life-history correlates which concur with their unique developmental trajectories [105]. Hence, it was necessary to include certain life history variables in the analysis to further understand how life-history characteristics potentially act as a filter [104, 106] for the production of large brains. Gestation length was chosen as it has received considerable attention and is thought to be of great importance in bypassing the constraints of precociality in mammals and facilitating brain growth [107]. Maximum lifespan was included as there is substantial support that encephalisation is correlated with extended longevity [104], especially in primates [108, 109]. The relationship found between brain size and lifespan is thought to be driven primarily by maternal investment, with subsequent correlations found between specific brain regions and developmental periods, reflecting this brain size-lifespan association (see [105, 110]. Ultimately encephalisation has been found to correlate with expansion of most developmental life history stages, including an extended reproductive lifespan [111]. Therefore, data on age at first reproduction, weaning and fertility (measured as number of offspring per year) were added to our dataset (see supplementary material for sources).

## Statistical analyses

**Brain transformations.**   Whole brain volumes were incorporated in analyses by simple incorporation of log ECV volume with log body mass included as a covariate. This method is often preferred over the use of residuals as variables in ecological datasets often covary thereby producing biased parameter estimates when calculating residuals [59]. Including body mass as a covariate in the model avoids this problem, controls for its effect on brain volume, as well as potentially controlling for any effects body mass may have on other variables included. Regional brain volumes were incorporated in analyses by simple incorporation of log ROB (rest of brain) volume. To calculate ROB volume for both the neocortex and cerebellum, a calculation was performed: whole brain volume minus the region volume of interest. This method has been previously implemented and proved useful in measuring relative regional

brain volumes (e.g., [91]). Further analyses were also conducted in order to test how uniform results were when using different brain size measures. The results of these analyses are displayed and discussed in the supplementary material.

**PGLS analysis.** All statistical analyses were performed using R 4.0.1, using the 'caper', 'ape' and 'geiger' packages. Phylogenetic generalised least-squares (PGLS) regression analysis was used to identify those variables influencing whole and regional brain evolution, whilst avoiding the problem of phylogenetic non-independence. This technique differs from standard generalised least squares analysis, as it uses knowledge of phylogenetic relationships or relatedness to produce estimates of the expected covariance across species [112]. Pagel's λ was estimated by maximum likelihood. The tree used for all phylogenetic analyses was that of Upham et al's [113]. All continuous variables, brain volumes and body mass were log transformed prior to analysis to satisfy the assumption of normality. Variance Inflation Factor (VIF) scores were used to check for the presence of multicollinearity, with almost all scores found to be below 5, and no scores above 7. There were no scores produced which highlighted concern, and thus, all socioecological and life-history variables were retained for analysis (see supplementary material).

**Model comparisons.** A series of PGLS models were implemented which varied in complexity, including 1) social, 2) ecological, 3) social and ecological, 4) life history and 5) variables of interest. Models one to four included all possible combinations of the selected variables; for example, the social model included i) group size, ii) social cohesion, iii) group size and social cohesion. BIC (Bayesian Information Criterion) values of each model were then compared [114]. As lower BIC values indicate the presence of better fitting, more parsimonious models, the model with the lowest BIC value was deemed to best explain the data, therefore considered preferrable and retained. BIC values were preferred over Akaike Information Criterion values because BIC resolves the problem of overfitting, by using a more conservative penalty for additional variables. Model number five was constructed using all variables previously highlighted of interest within the social, ecological, and life history models. As well as separating out proximate and ultimate causes of brain size evolution, this allowed us to compare the importance of social versus ecological models, constructing models that included those variables best explaining the data. Once computed, model five was compared alongside the previous models, and those found to have the lowest BIC value were then considered the '*best fit*' models, which in some cases represents a subset of models (simply, any model within dBIC<2 of the lowest model). This is because BIC values with a difference of between 2 and 6 indicate moderate evidence that the model with the lower BIC provides a relatively better model fit, whilst greater than 6 indicates strong evidence for improved fit.

## Results

### Primates

The results from PGLS analysis on the primate data are shown in Table 1. Almost all models were highly significant. For most models λ was close to one, indicative of a Brownian motion model of trait evolution; however, certain neocortex models stand in contrast to this, with λ equal to zero, implying the data have no phylogenetic structure [84]. The overall model section represents the different categories of PGLS models i.e., social, ecological. The preferred models section presents the model with the lowest BIC score within that respective category. For example, when investigating endocranial volume (with body mass), in the social category, the model with social cohesion produced the lowest score, whereas in the ecological category, the model with dietary breadth produced the lowest score.

When comparing BIC scores across all the models, combined models were preferred when investigating both whole and regional brain volumes (highlighted in bold), with significantly

**Table 1. Phylogenetic generalised least-squares (PGLS) regression analyses examining the effects of social, ecological and life-history variables[*] on primate whole and regional brain volumes.** Preferred models represent the '*best fit*' model (with the lowest BIC score) of the overall model category (i.e., social or ecological). The combined models represent the '*best fit*' model after running all combinations of the previous '*best fit*' models (models one to four). Boldness indicates the model(s) with the lowest BIC score across all models (dBIC<2).

| Brain input | Overall model | Preferred model | BIC score | *P*-value | λ | Adj. r² | Sample size (n) |
|---|---|---|---|---|---|---|---|
| **Endocranial volume** | Social | ECV ~ Mass + SC | -184.199 | <0.001 | 1 | 0.8774 | 83 |
| | Ecological | ECV ~ Mass + DB | -190.8458 | <0.001 | 1 | 0.8868 | 83 |
| | Social & Ecological | ECV ~ Mass + SC + DB | -192.0528 | <0.001 | 1 | 0.8929 | 83 |
| | Life History | ECV ~ Mass + GL + ML + WA | -201.2257 | <0.001 | 1 | 0.9079 | 83 |
| | **Combined** | **ECV ~ Mass + GS + DB + GL + ML + WA** | **-208.5244** | **<0.001** | **1** | **0.9222** | **83** |
| | All | ECV ~ Mass + GS + SC + D + DB + HV + HR + GL + ML + F + FR + WA | -183.9911 | <0.001 | 1 | 0.9207 | 83 |
| **Neocortex (ROB)** | Social | Neo ~ SC | 36.43372 | <0.05 | 0.991 | 0.08278 | 52 |
| | Ecological | Neo ~ D + HR | 20.04 | <0.001 | 0.843 | 0.481 | 52 |
| | Social & Ecological | Neo ~ SC + D + HR | 23.04369 | <0.001 | 0.866 | 0.4672 | 52 |
| | Life History | Neo ~ ML + WA | -9.507772 | <0.001 | 0 | 0.8602 | 52 |
| | **Combined** | **Neo ~ D + HR + ML + WA** | **-17.54041** | **<0.001** | **0** | **0.8984** | **52** |
| | All | Neo ~ GS + SC + D + DB + HV + HR + GL + ML + F + FR + WA | 9.397628 | <0.001 | 0 | 0.8818 | 52 |
| **Cerebellum (ROB)** | Social | Cere ~ SC | 26.55957 | <0.05 | 1 | 0.08632 | 49 |
| | Ecological | Cere ~ D + HR | 0.2775847 | <0.001 | 1 | 0.5238 | 49 |
| | Social & Ecological | Cere ~ SC + D + HR | 3.144599 | <0.001 | 1 | 0.5231 | 49 |
| | Life History | Cere ~ ML + WA | -17.40863 | <0.001 | 1 | 0.6485 | 49 |
| | **Combined** | **Cere ~ D + HR + ML + WA** | **-25.9437** | **<0.001** | **0.986** | **0.7631** | **49** |
| | All | Cere ~ GS + SC + D + DB + HV + HR + GL + ML + F + FR + WA | -10.45452 | <0.001 | 0.996 | 0.7699 | 49 |

[*]GS = Group size, SC = Social cohesion, D = Diet, DB = Dietary breadth, HV = Habitat variability, HR = Home range, GL = Gestation length, ML = Maximum longevity, F = Fertility, FR = Age at first reproduction, WA = Weaning age.

improved (equal or greater than two BIC units lower than another) BIC scores when combining variables indicated to be of importance in previous model iterations. When comparing the influence of ecology versus sociality, ecological models were found to be preferable to social models, evidenced by the presence of significantly improved BIC scores.

**Overall encephalisation.** The results of PGLS analysis on endocranial volume data are presented in Table 1, with the '*best fit*' models presented in Table 2. The variables which were indicated to be of importance and included in the '*best fit*' endocranial volume models were: group size, dietary breadth, gestation length, maximum lifespan and weaning age. Also present in the subset of '*best fit*' models were: social cohesion and home range. After accounting for phylogeny, both group size and social cohesion were found to be positively associated with ECV (P <0.05). Although, social cohesion failed to reach significance in certain model iterations (P = 0.06). In terms of the ecological variables, dietary breadth was consistently associated with ECV (P <0.001); however, home range size failed to reach significance (P = 0.08, 0.11). Three of the life-history variables were significantly associated with ECV: gestation length, maximum lifespan and weaning age (P <0.01).

**Regional brain volumes.** The results of PGLS analysis on the neocortex and cerebellum data are presented in Table 1, with the '*best fit*' models presented in Table 2. The variables which were indicated to be of importance and included within the '*best fit*' neocortex model were: diet, home range size, maximum lifespan and weaning age. After accounting for phylogeny, diet, specifically frugivory and omnivory were found to be negatively associated with

**Table 2. Phylogenetic generalised least-squares (PGLS) regression analyses examining the effects of social, ecological and life-history variables* on primate whole and regional brain volumes.** Preferred models represent all the '*best fit*' models for each brain input, which in most cases represents a subset of models (any model within dBIC<2 of the lowest model). This can include any category of model (i.e., social or combined), and is dependent on the BIC score produced. Boldness indicates <0.05.

| Brain input | Preferred models | BIC score | Predictor | Estimate | *t*-value | *P*-value |
|---|---|---|---|---|---|---|
| **Endocranial volume** | ECV ~ Mass + GS + DB + GL + ML + WA | -208.5244 | Intercept | -1.8599 | -6.6214 | **<0.001** |
| | | | LogMass | 0.5479 | 18.9909 | **<0.001** |
| | | | LogGS | 0.0432 | 2.1248 | **<0.05** |
| | | | DB | 0.0213 | 3.2392 | **<0.01** |
| | | | LogGL | 0.4021 | 2.8949 | **<0.01** |
| | | | LogML | 0.1488 | 3.0356 | **<0.01** |
| | | | LogWA | 0.1294 | 3.3570 | **<0.01** |
| | ECV ~ Mass + SC + DB + GL + ML + WA | <2 | Intercept | -1.8367 | -6.5280 | **<0.001** |
| | | | LogMass | 0.5463 | 18.8287 | **<0.001** |
| | | | SC | 0.0212 | 2.0765 | **<0.05** |
| | | | DB | 0.0233 | 3.5498 | **<0.001** |
| | | | LogGL | 0.3950 | 2.8406 | **<0.01** |
| | | | LogML | 0.1374 | 2.7985 | **<0.01** |
| | | | LogWA | 0.1257 | 3.2441 | **<0.01** |
| | ECV ~ Mass + DB + GL + ML + WA | <2 | Intercept | 0.2872 | -6.4578 | **<0.001** |
| | | | LogMass | 0.0293 | 18.9869 | **<0.001** |
| | | | DB | 0.0067 | 3.3586 | **<0.01** |
| | | | LogGL | 0.1420 | 2.7831 | **<0.01** |
| | | | LogML | 0.0501 | 2.8653 | **<0.01** |
| | | | LogWA | 0.0393 | 3.4476 | **<0.001** |
| | ECV ~ Mass + DB + HR + GL + ML + WA | <2 | Intercept | -1.8559 | -6.5533 | **<0.001** |
| | | | LogMass | 0.5387 | 17.7337 | **<0.001** |
| | | | DB | 0.0230 | 3.4826 | **<0.001** |
| | | | LogHR | 0.0178 | 1.7881 | 0.08 |
| | | | LogGL | 0.4195 | 2.9817 | **<0.01** |
| | | | LogML | 0.1383 | 2.7961 | **<0.01** |
| | | | LogWA | 0.1271 | 3.2575 | **<0.01** |
| | Mass + SC + DB + HR + GL + ML + WA | <2 | Intercept | -1.8391 | -6.6062 | **<0.001** |
| | | | LogMass | 0.5318 | 17.6895 | **<0.001** |
| | | | SC | 0.0196 | 1.9298 | 0.06 |
| | | | DB | 0.0237 | 3.6480 | **<0.001** |
| | | | LogHR | 0.0159 | 1.6222 | 0.11 |
| | | | LogGL | 0.4167 | 3.0146 | **<0.01** |
| | | | LogML | 0.1333 | 2.7384 | **<0.01** |
| | | | LogWA | 0.1190 | 3.0851 | **<0.01** |
| **Neocortex** | Neo ~ D + HR + ML + WA | -17.54041 | Intercept | 1.5482 | 6.0124 | **<0.001** |
| | | | DFrug | -0.1570 | -2.1200 | **<0.05** |
| | | | DOmni | -0.3093 | -3.9187 | **<0.001** |
| | | | LogHR | 0.1139 | 3.2303 | **<0.01** |
| | | | LogML | 0.6851 | 4.4548 | **<0.001** |
| | | | LogWA | 0.6482 | 6.4547 | **<0.001** |
| **Cerebellum** | Cere ~ D + HR + ML + WA | -25.9437 | Intercept | 2.3101 | 7.4158 | **<0.001** |
| | | | DFrug | -0.1131 | -1.5536 | 0.13 |
| | | | DOmni | -0.2645 | -3.0869 | **<0.01** |
| | | | LogHR | 0.1480 | 4.2338 | **<0.001** |

(*Continued*)

**Table 2.** (Continued)

| Brain input | Preferred models | BIC score | Predictor | Estimate | *t*-value | *P*-value |
|---|---|---|---|---|---|---|
| | | | LogML | 0.4402 | 3.0810 | **<0.01** |
| | | | LogWA | 0.5789 | 5.8047 | **<0.001** |
| | Cere ~ D + HR + GL + ML + WA | <2 | Intercept | 0.9767 | 1.2227 | 0.23 |
| | | | DFrug | -0.0762 | -1.0319 | 0.31 |
| | | | DOmni | -0.2336 | -2.7180 | **<0.01** |
| | | | LogHR | 0.1529 | 4.4768 | **<0.001** |
| | | | LogGL | 0.7857 | 1.8597 | 0.07 |
| | | | LogML | 0.3589 | 2.4562 | **<0.05** |
| | | | LogWA | 0.4390 | 3.6953 | **<0.001** |

*GS = Group size, SC = Social cohesion, D = Diet, DB = Dietary breadth, HV = Habitat variability, HR = Home range, GL = Gestation length, ML = Maximum longevity, F = Fertility, FR = Age at first reproduction, WA = Weaning age.

neocortex volume (P <0.05, P <0.001). This is the result produced when a folivorous diet is used as the baseline category, therefore the dietary category results produced here only demonstrates differences between these dietary groups (frugivory and omnivory) and folivory. Alongside these associations, home range size was positively correlated with neocortex volume (P <0.01). Similar to whole brain models, both maximum lifespan and weaning age were significantly associated with neocortex volume (P <0.001).

The variables which were indicated to be of importance and included in the '*best fit*' cerebellum models were: diet, home range size, maximum lifespan and weaning age. Also present within the subset of '*best fit*' models was: gestation length. After accounting for phylogeny, diet, specifically omnivory was found to be negatively associated with cerebellum volume (P <0.01). Frugivory failed to be significant (P = 0.13, P = 0.31). As above, this results when folivorous diet is used as the baseline category. Home range size was positively associated with cerebellum volume (P <0.001). Similar to previous life-history results, maximum lifespan and weaning age were significantly associated with cerebellum volume (P <0.01, P <0.001). Gestation length was close to being significantly correlated with cerebellum volume (P = 0.07).

## Carnivores

The results of PGLS analysis on the carnivore data are presented Table 3. Almost all models were highly significant. Lambda was not consistent between the models, ranging from one to zero across the dataset. The overall model section represents the different categories of PGLS models i.e., social, ecological. The preferred models section presents the model with the lowest BIC score within that respective category. In terms of the '*best fit*' models, those producing the lowest BIC score (or any score within dBIC<2 of the lowest model), there was no significant difference between life history and combined models (highlighted in bold), and thus the results of all these models are discussed below. When comparing the influence of ecology versus sociality, ecological models were found to be preferable to social models when investigating regional brain volumes, evidenced by the presence of significantly improved BIC scores. However, this was not the case in whole brain models, where there was no significant difference between the preferred social and ecological models.

**Overall encephalisation.** The results of PGLS analysis on endocranial volume data are presented in Table 3, with the '*best fit*' models shown in Table 4. The variables which were indicated to be of importance and included within the '*best fit*' endocranial volume models

**Table 3. Phylogenetic generalised least-squares (PGLS) regression analyses examining the effects of social, ecological and life-history variables* on carnivoran whole and regional brain volumes.** Preferred models represent the '*best fit*' model (with the lowest BIC score) of the overall model category (i.e., social or ecological). The combined models represent the '*best fit*' model after running all combinations of the previous '*best fit*' models (models one to four). Boldness indicates the model(s) with the lowest BIC score across all models (dBIC<2).

| Brain input | Overall model | Preferred model | BIC score | *P*-value | λ | Adj. r² | Sample size (n) |
|---|---|---|---|---|---|---|---|
| **Endocranial volume** | Social | ECV ~ Mass + GS | -137.3671 | <0.001 | 0.784 | 0.911 | 85 |
| | Ecological | ECV ~ Mass + HV | -138.8228 | <0.001 | 0.810 | 0.9102 | 85 |
| | Social & Ecological | ECV ~ Mass + GS + HV | -135.0748 | <0.001 | 0.814 | 0.9095 | 85 |
| | **Life History** | **ECV ~ Mass + F** | **-140.9778** | **<0.001** | **0.762** | **0.9166** | **85** |
| | **Combined** | **ECV ~ Mass + DB + F** | **-140.4778** | **<0.001** | **0.753** | **0.9201** | **85** |
| | All | ECV ~ Mass + GS + SC + D + DB + HV + HR + GL + ML + F + FR + WA | -106.9128 | <0.001 | 0.724 | 0.9221 | 85 |
| **Neocortex (ROB)** | Social | Neo ~ GS | 71.58854 | 0.06425 | 0.954 | 0.05726 | 44 |
| | Ecological | Neo ~ HR | 68.10774 | <0.01 | 0.334 | 0.196 | 44 |
| | Social & Ecological | Neo ~ GS + HR | 70.20444 | <0.01 | 0.400 | 0.1938 | 44 |
| | **Life History** | **Neo ~ FR** | **58.64386** | **<0.001** | **0.097** | **0.414** | **44** |
| | **Combined** | **Neo ~ HR + FR** | **59.78632** | **<0.001** | **0** | **0.48** | **44** |
| | All | Neo ~ GS + SC + D + DB + HV + HR + GL + ML + F + FR + WA | 87.42208 | <0.001 | 0 | 0.4546 | 44 |
| **Cerebellum (ROB)** | **Social** | Cere ~ GS | 35.60386 | 0.07056 | 1 | 0.06265 | 38 |
| | Ecological | Cere ~ HR | 20.3267 | <0.001 | 1 | 0.3729 | 38 |
| | Social & Ecological | Cere ~ GS + HR | 22.22221 | <0.001 | 1 | 0.3839 | 38 |
| | **Life History** | **Cere ~ GL + ML + FR** | **4.668459** | **<0.001** | **1** | **0.6369** | **38** |
| | **Combined** | **Cere ~ HR + GL + ML + FR** | **3.803654** | **<0.001** | **1** | **0.6677** | **38** |
| | All | Cere ~ GS + SC + D + DB + HV + HR + GL + ML + F + FR + WA | 28.10051 | <0.001 | 1 | 0.6135 | 38 |

*GS = Group size, SC = Social cohesion, D = Diet, DB = Dietary breadth, HV = Habitat variability, HR = Home range, GL = Gestation length, ML = Maximum longevity, F = Fertility, FR = Age at first reproduction, WA = Weaning age.

were: fertility, dietary breadth, maximum longevity and age at first reproduction. After accounting for phylogeny, fertility was found to be negatively associated with ECV (P <0.05), with this being the only variable significantly associated with endocranial volume. For example, dietary breadth was close to being negatively associated with ECV, but fell short of significance (P = 0.05). In addition, both maximum lifespan and age at first reproduction, failed to reach significance (P = 0.08, P = 0.10).

**Regional brain volumes.** The results of PGLS analysis on the neocortex and cerebellum data are presented in Table 3, with the '*best fit*' models shown in Table 4. The variables which were indicated to be of importance and included in the '*best fit*' neocortex models were: age at first reproduction, maximum lifespan and home range size. After accounting for phylogeny, age at first reproduction was found to be positively associated with neocortex (P <0.001), with this being the only variable significantly associated with neocortex volume. For example, home range size was close to being positively associated with neocortex volume, but fell short of significance (P = 0.07). In addition, maximum lifespan failed to reach significance (P = 0.19).

The variables which were indicated to be of importance and included within the '*best fit*' cerebellum models were: home range size, gestation length, maximum lifespan and age at first reproduction. Also present within the subset of '*best fit*' models were: different iterations of the previously mentioned variables and weaning age. After accounting for phylogeny, home range

**Table 4. Phylogenetic generalised least-squares (PGLS) regression analyses examining the effects of social, ecological and life-history variables\* on carnivoran whole and regional brain volumes.** Preferred models represent all the '*best fit*' models for each brain input, which in most cases represents a subset of models (any model within dBIC<2 of the lowest model). This can include any category of model (i.e., social or combined), and is dependent on the BIC score produced. Boldness indicates <0.05.

| Brain input | Preferred models | BIC score | Predictor | Estimate | *t*-value | *P*-value |
|---|---|---|---|---|---|---|
| **Endocranial volume** | ECV ~ Mass + F | -140.9778 | Intercept | -0.6057 | -5.3678 | **<0.001** |
| | | | LogMass | 0.5870 | 25.7757 | **<0.001** |
| | | | LogF | -0.1113 | -2.0993 | **<0.05** |
| | ECV ~ Mass + DB + F | <2 | Intercept | -0.5245 | -4.4263 | **<0.001** |
| | | | LogMass | 0.5810 | 25.6777 | **<0.001** |
| | | | DB | -0.0154 | -1.9622 | 0.05 |
| | | | LogF | -0.1318 | -2.4784 | **<0.05** |
| | ECV ~ Mass + ML | <2 | Intercept | -0.9083 | -7.0336 | **<0.001** |
| | | | LogMass | 0.5867 | 24.0699 | **<0.001** |
| | | | LogML | 0.1906 | 1.7925 | 0.08 |
| | ECV ~ Mass + FR | <2 | Intercept | -0.6513 | -6.0877 | **<0.001** |
| | | | LogMass | 0.5783 | 21.5774 | **<0.001** |
| | | | LogFR | 0.1145 | 1.6682 | 0.1 |
| **Neocortex** | Neo ~ FR | 58.64386 | Intercept | 4.0097 | 35.4993 | **<0.001** |
| | | | LogFR | 1.4150 | 5.6022 | **<0.001** |
| | Neo ~ ML + FR | <2 | Intercept | 2.8747 | 3.3575 | **<0.01** |
| | | | LogML | 0.9151 | 1.3334 | 0.19 |
| | | | LogFR | 1.0190 | 2.6229 | **<0.05** |
| | Neo ~ HR + FR | <2 | Intercept | 3.6343 | 17.222 | **<0.01** |
| | | | LogHR | 0.1437 | 1.856 | 0.07 |
| | | | LogFR | 1.0956 | 3.786 | **<0.001** |
| **Cerebellum** | Cere ~ HR + GL + ML + FR | 3.803654 | Intercept | 1.5075 | 1.8971 | 0.07 |
| | | | LogHR | 0.0753 | 2.0374 | **<0.05** |
| | | | LogGL | 0.8236 | 2.0974 | **<0.05** |
| | | | LogML | 0.9084 | 2.7665 | **<0.01** |
| | | | LogFR | 0.4524 | 2.1567 | **<0.05** |
| | Cere ~ GL + ML + FR | <2 | Intercept | 1.7089 | 2.0734 | **<0.05** |
| | | | LogGL | 0.7669 | 1.8730 | 0.07 |
| | | | LogML | 0.9706 | 2.8402 | **<0.01** |
| | | | LogFR | 0.6920 | 3.8113 | **<0.001** |
| | Cere ~ ML + FR | <2 | Intercept | 2.9664 | 5.9931 | **<0.001** |
| | | | LogML | 1.0852 | 3.1178 | **<0.01** |
| | | | LogFR | 0.8402 | 4.9662 | **<0.001** |
| | Cere ~ HR + ML + FR | <2 | Intercept | 2.8682 | 5.9347 | **<0.001** |
| | | | LogHR | 0.0702 | 1.8137 | 0.08 |
| | | | LogML | 1.0316 | 3.0414 | **<0.01** |
| | | | LogFR | 0.6336 | 3.1242 | **<0.01** |
| | Cere ~ ML + FR + WA | <2 | Intercept | 2.5812 | 4.7991 | **<0.001** |
| | | | LogML | 0.9485 | 2.7130 | **<0.05** |
| | | | LogFR | 0.7819 | 4.4666 | **<0.001** |
| | | | LogWA | 0.2815 | 1.6954 | 0.1 |

\*GS = Group size, SC = Social cohesion, D = Diet, DB = Dietary breadth, HV = Habitat variability, HR = Home range, GL = Gestation length, ML = Maximum longevity, F = Fertility, FR = Age at first reproduction, WA = Weaning age.

size was found to be significantly associated with cerebellum volume (P <0.05). Three of the life-history variables were significantly associated with cerebellum volume: gestation length, maximum lifespan and age at first reproduction (P <0.05, P <0.01, P <0.001). Although, home range size and gestation length failed to reach significance in certain model iterations (P = 0.08, P = 0.07). Weaning age also failed to reach significance (P = 0.10).

## Discussion

Applying robust statistical analyses, a recently updated phylogenetic tree, a comprehensive dataset and models of varying complexity, the correlates of brain size in primates and carnivores were reconsidered. Consistent associations were found between brain size and ecological variables in primates, thus highlighting the influence of ecology on encephalisation. However, support was also found for the prominent social brain hypothesis, specifically revealing evidence for a link between whole brain volumes and two measures of sociality. In carnivores, data suggest ecological variables shape brain size, suggesting alternative evolutionary patterns influencing carnivoran encephalisation. In both groups, life history variables appear crucial in counterbalancing the costs of producing and maintaining increased brain size, through extended developmental periods, reduced fertility and increased maximum lifespan.

### Primates

Here, consistent with current literature, robust correlations were found between brain size and ecological variables. The most prominent of these were diet related, with dietary categories or dietary breadth appearing in all '*best fit*' models, for both whole brain and regional brain data. These findings are similar to those of DeCasien et al., [20] and Powell et al. [14], who found stronger and more consistent associations with ecological variables than those related to the social environment. Akin to the result of DeCasien et al. [20], support was found for omnivory, as well as frugivory, as correlates of brain size. However, in contrast to the literature, here the correlations between regional brain volumes and dietary categories, were negatively correlated. This perhaps reflects both the need to sustain the energetic cost of brain tissue (highlighted by [115, 116]), as well as meeting the cognitive foraging challenges imposed by omnivorous and frugivorous diets [3]. In addition to the dietary categories, dietary breadth was significantly (positively) correlated with whole brain volumes, further reinforcing the proposition that diet influences brain size, whilst highlighting how useful this proxy can be in understanding how availability and variety of food sources can be important in setting the cognitive challenge. For example, MacLean et al. [50] also suggested dietary breadth to be an important ecological correlate, with greater cognitive flexibility allowing individuals to explore and exploit new food sources, as well as deploy extractive foraging techniques. Evidence for associations between regional brain volumes and home range size were also found, supporting the view of Powell et al. [14] in that certain dietary categories, such as frugivory, may covary with home range. Similar results were also found by Graber et al. [117].

In the past considerable support indicated that sociality was the major driver of encephalisation in primates. More recent works, however, contest this long-held viewpoint, failing to find support for a link between brain size and sociality measures [14, 19, 20, 50, 51]. Our findings, however, confirm support for the social brain hypothesis. Here, our models revealed evidence of a link between brain size and sociality in primates, potentially as a result of the model selection techniques used here which allowed the inclusion of multiple variables and because aspects of the social and ecological hypotheses are likely to covary. This association was present only in the whole brain '*best fit*' models, with both variables reaching significance, indicating both increasing social group size and varying levels of social cohesion are influencing brain size in primates.

Interestingly, use of the social cohesion proxy was often preferred when comparing models, thereby suggesting the use of this proxy is superior when testing multiple ecological and social variables simultaneously. The inference too is that there may be greater importance in relationship quality, over quantity, as suggested by past research into primate sociality and pair-bonds [34, 45, 49, 95, 118]. It is important to note however, that whilst there was support for this hypothesis, ecological models were preferrable over social ones and ecological variables appear to be more robust correlates of brain size when compared to measures of sociality (see [117]).

Consistent with the literature, support was found for correlations between life-history variables and brain size. As suggested within the developmental cost [110] and maternal energy [119] hypotheses, relationships found possibly reflect the developmental costs associated with growing large brains, which appear to be bypassed through extended developmental periods and increased maternal investment [120, 121]. Similarly, Powell et al. [105] found correlations between neocortex volume and gestation length, as well as cerebellum volume and juvenile period. The associations found here differ in terms of the specific regions involved, with methodological differences likely to underscore those differences in results. Powell et al., [105] for example, used body mass to control for allometric scaling of regional brain volumes whereas here the rest of brain technique was used, with this method also producing different results when we investigated regional brain volumes and the influence of diet. Despite these disparities, our results still support the theory as to why relatively large-brained mammals often exhibit slow maturation times and reduced fertility; thus, by increasing developmental periods and maternal investment, primates possess these slow life histories which ultimately facilitates the production of big brains. This therefore makes the 'extended parenting' association critical to the evolution of cognition [90, 120, 122, 123]. One mystery still left to solve, however, is the reasoning behind the association found here between brain size and maximum longevity. One proposition is that selection mechanisms work towards counterbalancing the costs of large brains in mammals with a longer reproductive lifespan [124], and thus, by extending the reproductive lifespan of a species, it counteracts the time and effort spent producing and maintaining large brains and aims to maximise the time species can spend producing young, which in turn have large brains. Whereas others propose the correlation is indirect and that a longer reproductive lifespan is a by-product of shifting developmental and maturation periods [105].

## Carnivores

Akin to the primate results, for carnivores, support is found for a link between regional brain volumes and home range size. This relationship reached significance in the cerebellum models, concurring with research suggesting this region is important for spatial memory processing [1, 125, 126]. Simply, larger home range sizes are thought to require the use of complex information about food location and distribution [9], which for example in carnivores, may represent the challenges of locating travelling herds of herbivores. Alongside this association, indicating spatial demands influence brain size in carnivores, dietary breadth was another ecological variable included in the '*best fit*' endocranial volume models. However, in contrast to the results of MacLean et al. [50] and Swanson et al. [19], the relationship between dietary breadth and brain size is negatively directed, suggesting greater dietary breadth is actually associated with smaller brain size in carnivores. This result could perhaps be a consequence of those species who are classified as obligate meat eaters, whose dietary breath is limited to one or two categories, thereby producing this negative correlation. Despite this, obligate meat-eating carnivores consume the highest caloric diet, which is thought to provide greater energy for producing large brains. This highlights how carnivores cannot simply be compared and likened to other mammalian orders, such as Primates, and suggests different evolutionary mechanisms at work

in carnivoran lineages. It is important to note, however, that this association, whilst close to, failed to reach significance (P = 0.05), suggesting this relationship is not a strong influence on brain size in carnivores.

Whilst previous work has suggested sociality plays a role in the evolution of brain size in carnivoran lineages [31, 33–35], here, we find no support for a link between measures of sociality and brain size in carnivores. Similarly, MacLean et al. [50], Benson-Amram et al. [127], and Swanson et al. [19], found no support for the social brain hypothesis in mammals. The contrasting results present in the literature could be due to the fact that sociality appears to be limited to a select few carnivore taxa, specifically social species from the families Hyaenidae, Procyonidae and Felidae [128]. This is suggested in the findings of Finarelli & Flynn [55], who identified that support for the SBH in Carnivora was dependent on data from Canidae, without which, no association is found. Thus, whilst sociality evidently plays an important role in primates, leading to complex, multi-faceted societies, this is less common in carnivore species, and therefore does not hold the same importance.

Consistent with the previously discussed primate results, associations were found between life-history variables and brain size in carnivores. Age at first reproduction, gestation length and maximum lifespan were all found to positively correlate with regional brain volumes, suggesting both an increase in developmental periods as well as an extension in reproductive lifespans. Additionally, findings are consistent with the expensive brain hypothesis [121], which proposes either an increase in energy turnover or a reduction in energy allocation is needed in order to meet the costs of increased brain size. This is seen here with a negative correlation between fertility and endocranial volume, suggesting a reduction in reproductive output. This, when paired with an increase in maternal investment and developmental periods, as suggested by the aforementioned results, bypasses the developmental constraints of producing a large brain through reduced fertility and slow maturation times.

### Whole versus regional brain volumes

Our study highlights the benefit of investigating both whole brain and regional brain volumes. Whole brain volumes are often more readily available for species and thus by choosing to use this brain measure it increases sample sizes and commensurate statistical power. In addition, it has been argued the neocortex comprises a large proportion of whole brain volume, making the two brain volumes closely related [34, 95]. However, it is possible the inclusion of specific brain regions may uncover further associations that were not significant or present before. This was the case here, where for primates, the home range association only became significant in the neocortex and cerebellum models, having not reached significance in endocranial volume models. Additionally, in carnivores, many of the life-history associations, for example age at first reproduction, only reached significance in the regional brain volume models. Therefore, without investigating specific brain regions, the influence of these associations would have been missed. In addition to this, the use of whole brain size does not necessarily allow the study of the ways in which different selective pressures act on different neural systems, as proposed by theories of mosaic evolution [5, 61]. This often makes it difficult to relate whole brain size to individual selection pressures [129]. By investigating specific brain regions, where brain data and the corresponding covariates are available, it allows the further analysis of how multiple functional systems can evolve in a mosaic fashion in response to different selection pressures.

### Conclusion

To conclude, the evidence presented here supports the proposition that ecological variables hold greater influence in determining brain size in primate lineages. However, critical support

is also found for the SBH in primates, confirming sociality does hold significance in encephalisation. Ecological variables, most notably home range size, appear to shape carnivoran brain size. Yet no support is found there for measures of sociality, indicating that sociality may not hold the same importance within that order. Life-history traits reveal evidence for the transition to slow life histories, which work toward facilitating the production of big brains and bypassing the cost of expensive brain tissue. Whilst data availability limits the application of comparative studies of brain evolution in many species, future studies should strive to integrate multiple variables, fully encompassing all the potential variables influencing brain size. In addition, where possible, researchers should investigate both whole brain and specific brain regions, as the inclusion of such may reveal further associations, capturing how different brain regions can evolve independently through varying selection pressures.

## Supporting information

**S1 File. Supplementary analyses.** This document includes information about the extra analyses conducted using different measures of brain size.
(DOCX)

**S2 File. Supplementary results tables.** This document includes all the supplementary results tables associated with the supplementary analyses.
(DOCX)

**S3 File. Supplementary BIC scores.** This excel file includes all the BIC scores used to conduct model comparisons during the main analyses.
(XLSX)

**S4 File. Additional BIC scores.** This excel file includes all the BIC scores used to conduct model comparisons during the extra analyses.
(XLSX)

**S5 File. Supporting data.** This excel file includes all the data used within the statistical analyses.
(XLSX)

**S6 File. VIF results.** This document includes all the VIF score results.
(DOCX)

**S7 File. Data collection sources.** This document includes all the data collection sources.
(DOCX)

**S8 File. R code.** This text file contains the R script used to conduct the statistical analyses.
(TXT)

**S9 File. Phylogenetic tree.** This file is the phylogenetic tree used during statistical analyses.
(NEX)

## Acknowledgments

We thank Alex DeCasien for help regarding encephalisation quotients and model comparison analyses. We are grateful to Eli Swanson for valuable discussions regarding PGLS analysis and for providing additional data analysis resources. Our thanks also to F. Sayol, O. Bertrand, V. Weisbecker, N. Emery, M. Tucker, D. Hinchcliffe, M. Olalla-Tárraga, S. Shultz, J. Gundry, C. O'Hara and C. Fauvelle. We are further grateful to the reviewers, whose thorough examinations led to a range of helpful suggestions that greatly assisted us in improving our ms.

## Author Contributions

**Conceptualization:** Helen Rebecca Chambers, Sean J. O'Hara.

**Data curation:** Helen Rebecca Chambers.

**Formal analysis:** Helen Rebecca Chambers, Sandra Andrea Heldstab.

**Investigation:** Helen Rebecca Chambers.

**Methodology:** Helen Rebecca Chambers, Sandra Andrea Heldstab, Sean J. O'Hara.

**Project administration:** Helen Rebecca Chambers.

**Supervision:** Sean J. O'Hara.

**Visualization:** Helen Rebecca Chambers, Sean J. O'Hara.

**Writing – original draft:** Helen Rebecca Chambers, Sean J. O'Hara.

**Writing – review & editing:** Helen Rebecca Chambers, Sandra Andrea Heldstab, Sean J. O'Hara.

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
