## [Decision Letter · Decision Letter 0]

9 Jun 2021

PONE-D-21-12399

Why big brains? A comparison of models for both primate and carnivore brain size evolution

PLOS ONE

Dear Dr. Chambers,

Thank you for submitting your manuscript to PLOS ONE. After careful consideration, we feel that it has merit but does not fully meet PLOS ONE’s publication criteria as it currently stands. Therefore, we invite you to submit a revised version of the manuscript that addresses the points raised during the review process.

I have now received two in-depth reviews of your submission. From my own perspective, I thought the manuscript was very well written, especially the introduction, and looks to add to the debate on drivers of brain size. Both reviewers agree that there is great merit to this work but have identified a series of issues that need to be addressed before the research could be published. Many of the comments ask for more clarity which should be easily dealt with. However, both also raise some statistical queries (e.g. reviewer 2's discussion of taking residuals from the regression line) and highlight some gaps in your references.

We look forward to receiving your revised manuscript.

Kind regards,

Adam Kane, PhD

Academic Editor

PLOS ONE

Journal Requirements:

2. Please include your tables as part of your main manuscript and remove the individual files. Please note that supplementary tables should remain uploaded as separate "supporting information" files.

Reviewers' comments:

Reviewer's Responses to Questions

**Comments to the Author**

1. Is the manuscript technically sound, and do the data support the conclusions?

Reviewer #1: Partly

Reviewer #2: Yes

2. Has the statistical analysis been performed appropriately and rigorously? 

Reviewer #1: No

Reviewer #2: Yes

3. Have the authors made all data underlying the findings in their manuscript fully available?

Reviewer #1: Yes

Reviewer #2: Yes

4. Is the manuscript presented in an intelligible fashion and written in standard English?

Reviewer #1: Yes

Reviewer #2: Yes

5. Review Comments to the Author

Reviewer #1: This is a well written manuscript that tests the correlates of brain size in primates and carnivores. The English is clear and free of typos. I do, however, have a number of concerns that need to be addressed before this manuscript can be considered acceptable for publication. I have listed these below, in no particular order.

1) I am not sure why this manuscript deals with just primates and carnivores. Why these two orders of mammals? Why not other orders such as rodents, lagomorphs, shrews, and bats? In fact, there are plenty of extensive datasets for these (and other) orders. For example, see Mace et al (1981) J. Zool. 193:333-354, which presents brain size data for 261 species of terrestrial small mammals, and Hutcheon et al. (2002) Brain Behavior Evolution 60:165-180 for 63 species of bats. I would have thought that a comparative approach across the entire class Mammalia would have been more fruitful than simply presenting data on primates and (incongruously) carnivores. The authors make no attempt to justify their selection of mammalian orders.

2) The literature cited is not representative of the field. A good deal of previous work has been omitted from this ms, including the two papers mentioned in (1) above, as well as Harvey et al. (1980) PNAS 77:4387-4389 (this paper explicitly deals with primate brain sizes). And there are many more papers that deal with ecological correlates of brain sizes that have not been mentioned.

3) Although the manuscript is generally well written, there are some sections that are difficult to interpret and/or to follow. This is particularly true for the Methods section, which is often ambiguous or at least incomplete. See below for where more detail is needed.

4) There is no definition of what is meant by the different brain volumes that are presented in the ms. For example, how was "endocranial brain volume" measured? And was it measured in the same way in the different papers where this information was extracted and collated? If not, then how can we be sure that we are comparing like with like?

5) The same comment applies to "neocortex" and "cerebellum" volumes.

6) Again, how was social cohesion measured? I can see that it was scored on a point system of 1 to 4, but what does it mean for a species to have a social cohesion of 1? or 2? etc. Without clearly defined explanatory variables, it is not possible to interpret the results of this study.

7) I found the ecological data simplistic and not at all credible. The authors will need to justify exactly what they mean by each of the ecological variables. And then, they will need to convince the reader that the ecological data are actually meaningful. I am happy to include "diet" (although "frugivore" or "omnivore" are diet categories rather than strictly speaking diet itself (and the authors actually refer to diet categories, but they don't explicitly make the distinction). But what do they mean by diet breadth? According to their definition it is: "dietary breadth was also used, estimated using the total number of food sources used by a species". But what are these "food sources"? Are they the number of species of plants/animals taken? If so, an insectivorous species will by definition have a wider breadth than a carnivorous one (because there are more species of insects than vertebrates). If "sources" refers to something else, then what is it? And then, once the definition has been clearly stated, how can we be sure that the different studies have scored "number of food sources" in the same way?

8) I have even more issue with the number of habitats used by a species. Wider ranging species will use a greater number of habitats, so why didn't the authors correct for this? Or simply use distributional range size instead of number of habitats?

9) The authors do not mention where they get their home range sizes from in the ms (although these are clearly mentioned in the supplementary material). I find it hard to believe that the various range sizes compiled by numerous authors will be directly comparable due to differences in techniques used to estimate home range. Furthermore, there is enormous amount of variation in home range size, which is partly (and only partly) attributable to sex and age. Using a single metric is hardly informative or convincing.

10) Statistical analysis. This entire section (lines 218 to 239) needs to be reworked and more detail provided. And unambiguous statements rephrased. I will make just a few examples (but these are not the only problems).

11) Lines 219-220 "using residuals from a regression line". Regression of what on what? And exactly using what regression? Simple linear regression e.g. lm()? On log transformed or untransformed data?

12) What is the encephalisation quotient and how was it calculated? In fact, the equation is presented a bit further down, so perhaps the authors just need to refer to this e.g. say something like "see below for equation".

13) Line 220. "The former method is often preferred...". But you can't use "former" when there are three methods presented. "Former" and "latter" can only be used when comparing two things.

14) Line 226. "...therefore we considered it prudent to use both methods in the analyses...". Which two methods are being referred to? Because the authors have mentioned three methods (which have even been numbered).

15) Please provide a basic description of "Phylogenetic generalised least-squares regression analysis" and how it differs from typical GLMs.

16) VIF was used to check for collinearity (which is good). But what does it mean "almost all scores" were below 5. Which variables were autocorrelated? And were any removed from the analyses, as a result of this?

17) Possible limitations. I find this paragraph difficult to accept. The authors are well aware that any models with AICs within 2 points are not "statistically different". Then how can they justify their approach? To me, this is the weakest aspect of the ms, because it affects all of their interpretations. There must be better ways of dealing with this. For example, list all competing models, and then count the number of times a particular variable (e.g. social cohesion) appears in the top models? This may make the results much more difficult to interpret, but this may be because there really is no simple and easy answer to the question that they are asking. Simplifying a complex problem with incorrect statistics is not acceptable.

I would like to see these concerns dealt with before the manuscript is accepted in this journal.

Reviewer #2: Please see attached.

6. PLOS authors have the option to publish the peer review history of their article (what does this mean?). If published, this will include your full peer review and any attached files.

Reviewer #1: No

Reviewer #2: **Yes: **Alex R. DeCasien

---

## [Author Response · Author response to Decision Letter 0]

31 Aug 2021

Reviewer #1

1) I am not sure why this manuscript deals with just primates and carnivores. Why these two orders of mammals? Why not other orders such as rodents, lagomorphs, shrews, and bats? In fact, there are plenty of extensive datasets for these (and other) orders. For example, see Mace et al (1981) J. Zool. 193:333-354, which presents brain size data for 261 species of terrestrial small mammals, and Hutcheon et al. (2002) Brain Behavior Evolution 60:165-180 for 63 species of bats. I would have thought that a comparative approach across the entire class Mammalia would have been more fruitful than simply presenting data on primates and (incongruously) carnivores. The authors make no attempt to justify their selection of mammalian orders.

Whilst we understand that brain data are available for more species than which were included within the manuscript, we wanted to run analyses on a complete dataset with all covariates available for all species, as this enabled more robust analyses, especially when conducting model comparisons. We could access all the required covariates for primates and carnivores, which governed our choice. In addition, in efforts to address the current confusion within the field regarding the proposed selection pressures responsible for increased brain size, we chose to use both primate and carnivore data as these two groups have received considerable attention, and thus by drawing clarity within these two groups, further groups can be studied using more appropriate methods/procedures. We have added wording to emphasise our reasoning for this choice. 

2) The literature cited is not representative of the field. A good deal of previous work has been omitted from this ms, including the two papers mentioned in (1) above, as well as Harvey et al. (1980) PNAS 77:4387-4389 (this paper explicitly deals with primate brain sizes). And there are many more papers that deal with ecological correlates of brain sizes that have not been mentioned.

Additional citations have been added. 

3) Although the manuscript is generally well written, there are some sections that are difficult to interpret and/or to follow. This is particularly true for the Methods section, which is often ambiguous or at least incomplete. See below for where more detail is needed.

Wording has been rephrased for added clarity. 

4) There is no definition of what is meant by the different brain volumes that are presented in the ms. For example, how was "endocranial brain volume" measured? And was it measured in the same way in the different papers where this information was extracted and collated? If not, then how can we be sure that we are comparing like with like?

5) The same comment applies to "neocortex" and "cerebellum" volumes.

Definitions have been added for endocranial and regional brain volumes. When sourcing all whole and regional brain volumes these measurement methods were considered to ensure the data was comparable. In terms of the ECV data, sources were checked for comparability and common measurement techniques were found between studies. We further tried to minimise the risk of this problem by sourcing data from whole datasets e.g., DeCasien et al., 2019 where the information has been weighted to account for multiple methods. However, this was more difficult with the carnivore data where regional brain volume data was tricky to source. 

6) Again, how was social cohesion measured? I can see that it was scored on a point system of 1 to 4, but what does it mean for a species to have a social cohesion of 1? or 2? etc.

Definition revised for greater clarity. 

7) I found the ecological data simplistic and not at all credible. The authors will need to justify exactly what they mean by each of the ecological variables. And then, they will need to convince the reader that the ecological data are actually meaningful. I am happy to include "diet" (although "frugivore" or "omnivore" are diet categories rather than strictly speaking diet itself (and the authors actually refer to diet categories, but they don't explicitly make the distinction). But what do they mean by diet breadth? According to their definition it is: "dietary breadth was also used, estimated using the total number of food sources used by a species". But what are these "food sources"? Are they the number of species of plants/animals taken? If so, an insectivorous species will by definition have a wider breadth than a carnivorous one (because there are more species of insects than vertebrates). If "sources" refers to something else, then what is it? And then, once the definition has been clearly stated, how can we be sure that the different studies have scored "number of food sources" in the same way?

Definitions of dietary categories and dietary breadth revised for greater clarity. All dietary breadth data was taken from one source: Wilman et al., (2014) and is referred to in the manuscript. 

8) I have even more issue with the number of habitats used by a species. Wider ranging species will use a greater number of habitats, so why didn't the authors correct for this? Or simply use distributional range size instead of number of habitats?

Whilst we understand and appreciate this point, it does not always follow that wider ranging species will always use a greater number of habitats. One species may have a large home range size but may only move within the same habitat type. What we instead aim to look at here is whether the type of habitat matters, thus, do species which navigate and confront multiple habitat types, have larger brains than those which only move within one or two habitat types? Or vice versa? We also use home range size to proxy habitat use. 

9) The authors do not mention where they get their home range sizes from in the ms (although these are clearly mentioned in the supplementary material). I find it hard to believe that the various range sizes compiled by numerous authors will be directly comparable due to differences in techniques used to estimate home range. Furthermore, there is enormous amount of variation in home range size, which is partly (and only partly) attributable to sex and age. Using a single metric is hardly informative or convincing.

We did not want to mention the citations specifically within the manuscript due to the high number of citations. We agree with this point about transferability of the methods used to measure home range size. We did our best to reduce the number of sources due to this problem, however, due to limited data availability, the only way to retrieve home range size for all species was to use data from multiple studies. To minimise the issue highlighted, we chose to use hectares to measure home range size as this was the most prevalent method found. We converted all home range data collected to this metric. We agree a single metric is not always useful, which is why we used both habitat variability and home range size to proxy habitat use. 

10) Statistical analysis. This entire section (lines 218 to 239) needs to be reworked and more detail provided. And unambiguous statements rephrased. I will make just a few examples (but these are not the only problems).

Wording has been rephrased for clarity. 

11) Lines 219-220 "using residuals from a regression line". Regression of what on what? And exactly using what regression? Simple linear regression e.g. lm()? On log transformed or untransformed data?

Phrase removed as this aspect has been moved to supplementary methods. This regression analysis is discussed in full within that document… “Phylogenetic generalised least-squares regression analysis (PGLS) was used to regress log brain volume against log body mass”. 

12) What is the encephalisation quotient and how was it calculated? In fact, the equation is presented a bit further down, so perhaps the authors just need to refer to this e.g. say something like "see below for equation".

Definition revised for greater clarity. This aspect – as mentioned above – has been moved to the supplementary methods. 

13) Line 220. "The former method is often preferred...". But you can't use "former" when there are three methods presented. "Former" and "latter" can only be used when comparing two things.

Thank you for highlighting. Phrase removed. 

14) Line 226. "...therefore we considered it prudent to use both methods in the analyses...". Which two methods are being referred to? Because the authors have mentioned three methods (which have even been numbered).

Phrase removed. 

15) Please provide a basic description of "Phylogenetic generalised least-squares regression analysis" and how it differs from typical GLMs.

Definition revised to provide greater clarity. 

16) VIF was used to check for collinearity (which is good). But what does it mean "almost all scores" were below 5. Which variables were autocorrelated? And were any removed from the analyses, as a result of this?

Almost all VIF scores produced were below 5, however there were a few outliers. For example, body mass and weaning age produced scores of 7.25 and 5.93, when inputted into the primate endocranial model. Whilst moderately high, we chose to retain all variables within the statistical models, as the scores were only found in a few models and were still considerably low. Thus, no variables were removed from the analyses. VIF scores were also checked when rerunning analyses, specifically when using the ‘rest of brain’ regional volume technique, with no scores produced of concern. 

This sentence has been updated to provide greater clarity. 

17) Possible limitations. I find this paragraph difficult to accept. The authors are well aware that any models with AICs within 2 points are not "statistically different". Then how can they justify their approach? To me, this is the weakest aspect of the ms, because it affects all of their interpretations. There must be better ways of dealing with this. For example, list all competing models, and then count the number of times a particular variable (e.g. social cohesion) appears in the top models? This may make the results much more difficult to interpret, but this may be because there really is no simple and easy answer to the question that they are asking. Simplifying a complex problem with incorrect statistics is not acceptable.

We appreciate this comment. We agree this was a weak point in the analyses. To address this highlighted shortcoming, rather than just choosing the model with the absolute lowest score, we have now adopted the approach of presenting and discussing the results of all the ‘best fit’ models, which usually included a subset of models (simply, all the models within 2 points of the absolute lowest model). We have also rerun the analysis using BIC rather than AIC, in acknowledgement of this scoring system being more conservative. 

Reviewer #2

• Line 33: See my comment in the Discussion section on the use of “counterbalancing”.

Wording rephrased. 

• There is a critical part currently missing this section, which is an explicit discussion of how this study is different from the many previous analyses of brain ~ socioecology relationships (e.g., inclusion of more variables, updated phylogeny, higher individual/species sample sizes)?

Thank you for this comment, we agree this was lacking in the manuscript. Introduction has been updated with this discussion. 

• Line 75: The importance of pair-bondedness to brain size evolution was also discussed in other papers, which should be cited here (Schillaci 2006, 2008; MacLean et al. 2009).

• Line 83: This reference is only for carnivores – please add a reference for primates. 

Citations added. 

• Paragraph starting with Line 90:

o I think a discussion of issues with relative brain size measures is important, however, I don’t think it warrants using measures that have been previously established as inappropriate (i.e., residuals, EQ).

• Lines 141-144: Again, it is unnecessary to include analyses using EQ or brain size residuals.

• Lines 218-220: Again, it is unnecessary to include analyses using EQ or brain size residuals.

• Paragraph starting with Line 467: As previously mentioned, previous studies have demonstrated that the use of EQ or residuals is inappropriate, so I think this paragraph and the relevant results are unnecessary and make the overall findings harder to follow. 

We appreciate that these methods have previously been suggested to be inappropriate for measuring the relationship between brain size and body mass. We feel it is necessary to further address this problem, however, especially considering we are using updated data, updated statistical analysis, more variables and an updated phylogenetic tree. After considering this point, we decided to move the results produced using the methods of concern (i.e., residuals, EQ) to the supplementary material and these will no longer be discussed in the main manuscript. This moves the focus away from those methods, but still allows the comparison between methods which may be useful to some readers. 

o The findings from the most recent study on brain ~ body size evolution (Smears et al. 2021) should be considered/discussed here. 

o Freckleton’s (2009) “seven deadly sins of comparative analysis” should be mentioned here, as it includes a discussion on why it is inappropriate to use residuals as outcome variables in regression models.

o Lines 105-107 – Papers on lag between primate brain and body size should be mentioned here (e.g., Deaner and Nunn 1999).

Thank you. Citations added. 

o Line 108: It is unclear what “over statistically controlled methods” means here.

Wording rephrased. 

o Line 109: How and why does van Schaik et al. (2021) specifically demonstrate that EQ is inappropriate? The authors should elaborate a bit here.

Some elaboration has been added, as recommended. 

• Paragraph starting with Line 111: 

o How would social and ecological variables specifically relate to neocortical and cerebellar functions? 

• Increased brain size is the result of selection on specific abilities and related neural systems. Accordingly, at some point in this Introduction, I would appreciate a brief but explicit discussion of this (e.g., why might frugivory require greater visual information processing? Given that a large proportion of the brain is neocortex, and a large proportion of the neocortex is comprised of visual information processing areas, might this explain the link between something like frugivory and overall brain size?)

These points are now discussed. 

o I think it would be appropriate to discuss Powell et al. (2019) here (currently only mentioned in the Discussion).

Powell et al., (2019) has been discussed further in the methods section. 

• Line 126: What kind of “models”?

Sentence has been elaborated upon. 

• Line 155: Please add sample sizes for the neocortex and cerebellum.

Sample sizes updated. 

• Lines 157-161: This is Introduction material and should be removed from the Methods.

• Paragraph starting with Line 163: It might be useful to include some of this in the Introduction, since readers have any background surrounding issues with various “social complexity” measures.

• All descriptions of the links between socioecological variables and selection for cognitive abilities would be more appropriate in the Introduction.

These sections have been moved to the introduction. 

• Lines 171-174: What were levels 2 and 3? How were pairbonded species or those that only sleep in pairs categorized? These levels need more explanation, especially since this “social cohesion” proxy was included in many best fit models in the Results.

Agreed. Definition revised for greater clarity. 

• Lines 196-197: Diet imposes both temporal and spatial cognitive demands, so I suggest re-wording this.

• Lines 200-203: The authors appear to be suggesting that certain life history variables are drivers of evolutionary changes in brain size. I suggest altering the language here to mimic that in Lines 421-424.

Sentences rephrased for clarity. 

• Paragraph staring with Line 200: This section is missing a discussion of ideas that the relationship between brain size and lifespan is driven by maternal investment and between specific brain regions and developmental periods (see e.g., Barton et al. 2011; Powell et al. 2019)

This point has been discussed. 

• Lines 238-239: Why was body mass used as the covariate for the neocortex and cerebellum models? Many other papers have used brain size (with the brain region of interest removed) or medulla size as a covariate. This decision should be justified in the text or analyses should be re-run using a brain size measure.

Thank you for this comment, we agree that this method needed to be altered. Neocortex and cerebellum size were recalculated using endocranial volume minus the brain region of interest. Analyses were re-run using this brain size measure. The method (brain transformations) section has been updated to reflect this change. 

• Model comparisons section: 

o This section as written is unclear – were the best fit models within Models 1-4 first identified, and then combined to make Model 5? 

o In any case, I do not think this approach is appropriate since it may, in some cases, force the inclusion of low information variables into the “combined” model. It would be more appropriate to create models that include all combinations of all predictor variables, compare these models using information criterion (I suggest using BIC since it is more conservative), and then select the best fit model or subset of models (e.g., all models with dBIC<2) to present detailed results.

Models one to four contained all combinations of the predictor variables, specifically looking at 1) social, 2) ecological, 3) social & ecological and 4) life history. Then usually models 3 and 4 were combined to determine whether incorporating the models together produced a better information criterion score. I say usually because sometimes incorporating social variables did not improve the score, therefore models 2 and 4 were combined instead. This combined model was also compared against a model including all variables together. We chose to use this ‘combined’ model because it would take too much time to try every combination of the 11 variables, therefore we thought by combining best fit models, this would bypass this problem and produce superior models. We appreciate your comment about the inclusion of low information variables, and it is definitely something we considered. After your suggestion, to better address the issue, the analyses have been re-run using BIC instead of AIC, due to the fact it is more conservative and would reduce the likelihood of low information variables being included. We also chose to present the results of the ‘best fit’ models, which was usually a subset of models (presenting all models within dBIC<2 of the absolute lowest model). 

• Lines 260-261: The meaning of “presently, and subsequently” is unclear.

Phrase removed for clarity. 

• This section is a bit difficult to follow as written. I suggest, within each section, more clearing separating/identifying the different groups of results. I think it would be most appropriate to first discuss results using the information criterion (i.e., tell the readers which variables are included in the best fit models) and then the frequentist results (i.e., tell the readers which coefficient estimates within the best fit model are “significant” and the direction of the relationship)?

Thank you for this comment, we agree and the results section has been rewritten to allow greater clarity. 

• Table 2: The diet category results (DFrug, DOmni) only demonstrate differences between these dietary groups (frugivory and omnivory) and folivory. This needs to be explicitly stated in the relevant areas of the results section. In addition, models should be run with the levels switched so that potential differences between frugivory and omnivory can also be tested. 

Thank you for this comment, we agree that this needed highlighting. This has now been explicitly stated in the primate results section. In addition, as suggested, models were run with the levels switched, to identify any potential differences between frugivory and omnivory. This was checked on all ‘best fit’ models where diet was included, thus, on both the primate neocortex and cerebellum combined models. To do this, primate regional volume data was used, with linear regression models implemented, using the same combination of variables seen in the combined models (Neo ~ D + HR + ML + WA, Cere ~ D + HR + ML + WA). 

Just included for your information…

Looking at primate neocortex data, when folivory was used as the baseline, negative significant associations were found with both omnivory and frugivory. However, when frugivory was used as the baseline, a positive association was found with folivory, whereas a negative association was found with omnivory. When omnivory was used as the baseline, positive associations were found with both frugivory and folivory. Thus, folivores appear to have larger neocortex volumes when compared to those with frugivorous and omnivorous diets, and this statement holds when the levels are switched (frugivorous and omnivorous species have smaller neocortex volumes when compared to those with a folivorous diet). However, frugivores appear to have larger neocortex volumes when compared to omnivores, and again, this statement holds when the levels are switched (omnivorous species have smaller neocortex sizes when compared to frugivorous species). 

Looking at primate cerebellum data, the results are similar; both folivorous and frugivorous species appear to have larger cerebellar volumes when compared to those with an omnivorous diet, with this statement holding when the levels are switched (omnivorous species have smaller cerebellum volumes when compared to those with folivorous and frugivorous diets). However, there appears to be no discernible difference between folivorous and frugivorous species in terms of cerebellum volume. 

• Lines 287-288 and 303-304: Table 2 includes results from best fit models only – it would be appropriate to also mention Table 1.

Table 1 has also been mentioned. 

• Lines 288-289: Diet is not included in the best fit model for ECV in Table 1, so I am a bit confused about the claim that diet is positively associated with all brain measures.

What we meant by this sentence was that diet as a whole (dietary categories or dietary breadth) was associated with all brain measures. We agree this should have been better worded. This sentence has been removed, however, following the recommendation to no longer discuss the different brain measures in the main manuscript. 

• Paragraph starting in Line 345: The home range results for the neocortex are not mentioned.

Thank you for pointing this out. We have now ensured all results are now appropriately discussed. 

• Lines 383-385: The finding that habitat variability is negatively correlated with relative brain size should be discussed in terms of previous work demonstrating a negative impact of seasonality on brain size (e.g., van Woerden et al. 2010).

This correlation is no longer found after rerunning statistical analyses so has been removed. 

• Lines 409-410: This is not true. Powell et al. (2019) found correlations between specific brain regions (neocortex) and gestation length. Other regions were correlated with other developmental periods (e.g., cerebellum and juvenile period).

Sentence updated to reflect this point. 

• Line 421: What does “counterbalance” mean? It sounds as if animals are actively participating in the evolution of these traits. Can the authors elaborate on how specific selection mechanisms would drive this “counterbalancing”?

Sentence updated to reflect this point. 

• Lines 426-427: This sentence makes it seem that diet category is included in the best fit models for carnivores, which is not the case. I suggest removing the sentence.

Sentence removed as recommended. 

• Lines 443-446: Sociality is not included in any of the best fit models of relative brain size, so this sentence is misleading as written.

Sentence changed following reanalysis of data. 

• Lines 445-457: I would remove this sentence since the cerebellum is showing opposite trends across groups.

Sentence removed.

---

## [Decision Letter · Decision Letter 1]

27 Oct 2021

PONE-D-21-12399R1Why big brains? A comparison of models for both primate and carnivore brain size evolutionPLOS ONE

Dear Dr. Chambers,

Thank you for submitting your manuscript to PLOS ONE. After careful consideration, we feel that it has merit but does not fully meet PLOS ONE’s publication criteria as it currently stands. Therefore, we invite you to submit a revised version of the manuscript that addresses the points raised during the review process.

We look forward to receiving your revised manuscript.

Kind regards,

Adam Kane, PhD

Academic Editor

PLOS ONE

Journal Requirements:

Additional Editor Comments (if provided):

The previous reviewers have gone through the updated draft and both recognize your extensive revisions in this version. This next batch of comments are relatively minor though I do strongly agree with them that you need to spend more time detailing what is shown in your tables. I'd also advocate for including the model coefficients rather than just t-values and p-values.

Reviewers' comments:

Reviewer's Responses to Questions

**Comments to the Author**

1. If the authors have adequately addressed your comments raised in a previous round of review and you feel that this manuscript is now acceptable for publication, you may indicate that here to bypass the “Comments to the Author” section, enter your conflict of interest statement in the “Confidential to Editor” section, and submit your "Accept" recommendation.

Reviewer #1: All comments have been addressed

Reviewer #2: (No Response)

2. Is the manuscript technically sound, and do the data support the conclusions?

Reviewer #1: Yes

Reviewer #2: Yes

3. Has the statistical analysis been performed appropriately and rigorously? 

Reviewer #1: Yes

Reviewer #2: Yes

4. Have the authors made all data underlying the findings in their manuscript fully available?

Reviewer #1: Yes

Reviewer #2: Yes

5. Is the manuscript presented in an intelligible fashion and written in standard English?

Reviewer #1: Yes

Reviewer #2: Yes

6. Review Comments to the Author

Reviewer #1: The authors have made extensive changes to their manuscript based on my prior comments. I am now happy to accept this manuscript for publication. However, there are a few minor issues that still need to be addressed (please see comments inserted into the attached PDF). I do not need to see a revision of this manuscript.

Reviewer #2: Comments to the Author (attached and below)

Review of “Why big brains? A comparison of models for both primate and carnivore brain size evolution” (PONE-D-21-12399_R1)

The authors have thoughtfully addressed my previous comments. Accordingly, I suggest this manuscript is published following minor revisions (outlined by section below).

Introduction

• Lines 100-105: The discussion of Smaers and colleagues’ work on different evolutionary paths to relative brain size needs a bit of reorienting. As written, it sounds as if their work represents justification against using EQ and towards using another measure of relative brain size. However, they interpret their findings to mean that relative brain size is not likely to always reflect selection on cognition, and that comparisons of this measure across species with different evolutionary histories do not address this.

• Line 147: I would add that research focus on primate evolution has also resulted from anthropocentrism.

Methods

• It might be useful to note that the different groups models run separate proximate (developmental) versus ultimate (ecological, social) causes of brain size evolution.

• VIF values of 5 correspond to R2 = 0.8 (which seems high) – were there many models approaching this VIF value?

Discussion

• Reasons why some of the results presented here contradict those from other recent studies (e.g., neocortex size predicted by gestation length in Powell et al. 2019) should be elaborated upon here.

Tables

• The legends should be more descriptive/comprehensive (e.g., only best fit models shown, how combinations derived, etc).

• I suggest removing the asterisks denoting level of significance in Table 2.

• Please note in the legends that boldness indicates p<0.05.

o Accordingly, the intercept and mass values should also be in bold.

• Where is ROB for the neocortex and cerebellum models? Was it not included (in contradiction to the methods) or was it accidentally omitted from the table)?

Supplement

• Table S3: It is unclear why the combined models do not represent all combinations for the social/ecological/life history best fit models (dBIC<2) – are only the combined models with dBIC<2 shown? This needs to be clarified somewhere.

I hope that these comments are useful in revising your manuscript.

Sincerely,

Alex DeCasien

7. PLOS authors have the option to publish the peer review history of their article (what does this mean?). If published, this will include your full peer review and any attached files.

Reviewer #1: No

Reviewer #2: No

---

## [Author Response · Author response to Decision Letter 1]

4 Nov 2021

Additional Editor Comments (if provided):

The previous reviewers have gone through the updated draft and both recognize your extensive revisions in this version. This next batch of comments are relatively minor though I do strongly agree with them that you need to spend more time detailing what is shown in your tables. I'd also advocate for including the model coefficients rather than just t-values and p-values.

Thank you for your comment about model coefficients; we agree that this inclusion could add to our tables. Model coefficient estimates have now been added to all appropriate tables. 

Reviewer #1: (reviewer comments are pictures in word doc)

Sentence altered. 

Thank you for highlighting this, we agree this was something that needed acknowledging. A sentence has been added highlighting this and briefly justifying our use of these techniques. 

Thank you for this comment but, as you acknowledge, it’s not necessary to refer the reader here as it’s covered elsewhere. 

To provide greater clarity this has been altered from P to P-value in all tables. 

Thank you for highlighting this. Table 1 and 3 have been discussed more fully in the results text. 

Thank you for highlighting this, you are correct. The bold row represents the model with the lowest BIC score. For all carnivore results both life history and combined models should have been highlighted (dBIC <2), this has been clarified. 

Reviewer #2:

Introduction

• Lines 100-105: The discussion of Smaers and colleagues’ work on different evolutionary paths to relative brain size needs a bit of reorienting. As written, it sounds as if their work represents justification against using EQ and towards using another measure of relative brain size. However, they interpret their findings to mean that relative brain size is not likely to always reflect selection on cognition, and that comparisons of this measure across species with different evolutionary histories do not address this.

Thank you for this comment, we agree the phrasing needed altering. The discussion of Smaers and colleagues’ work has been now been reorientated. 

• Line 147: I would add that research focus on primate evolution has also resulted from anthropocentrism.

Comment added. 

Methods

• It might be useful to note that the different groups models run separate proximate (developmental) versus ultimate (ecological, social) causes of brain size evolution.

Thank you for highlighting this, it is definitely something we should have mentioned. A sentence has been added. 

• VIF values of 5 correspond to R2 = 0.8 (which seems high) – were there many models approaching this VIF value?

We only had three occurrences of scores around 5. Two in the primate endocranial volume model, with body mass producing 6.5 and weaning age producing 6, and the last in the carnivore cerebellum model, with age at first reproduction producing 5. All other scores were below 3 (see supplementary material for all VIF scores). Whilst we appreciate these scores are of slight concern, we felt it necessary and appropriate to retain all variables for further analysis because the (relatively) high scores were still only present in a few models. 

Discussion

• Reasons why some of the results presented here contradict those from other recent studies (e.g., neocortex size predicted by gestation length in Powell et al. 2019) should be elaborated upon here.

Discussions on the potential reasoning behind the contrasting results have been added. Just to note, in order to incorporate this discussion, it was necessary to re-work the sentences and structure of the paragraph. 

Tables

• The legends should be more descriptive/comprehensive (e.g., only best fit models shown, how combinations derived, etc).

Thank you for highlighting this, we agree the legends could benefit from having more extensive descriptions. All table legends have been updated. 

• I suggest removing the asterisks denoting level of significance in Table 2.

• Please note in the legends that boldness indicates p<0.05.

o Accordingly, the intercept and mass values should also be in bold.

All points here have been completed. 

• Where is ROB for the neocortex and cerebellum models? Was it not included (in contradiction to the methods) or was it accidentally omitted from the table)?

The neocortex and cerebellum sections in the tables represent the results produced when using the ROB technique. We appreciate this was not obvious, so this has been highlighted in tables 1 and 3. 

Supplement

• Table S3: It is unclear why the combined models do not represent all combinations for the social/ecological/life history best fit models (dBIC<2) – are only the combined models with dBIC<2 shown? This needs to be clarified somewhere.

Thank you for highlighting this, we had neglected to include all the combined models. We have now rectified this, and combined models now display all combinations of the social/ecological/life history best fit models (seen in the BIC score excel files).

---

## [Editor Report · Decision Letter 2]

25 Nov 2021

Why big brains? A comparison of models for both primate and carnivore brain size evolution

PONE-D-21-12399R2

Dear Dr. Chambers,

We’re pleased to inform you that your manuscript has been judged scientifically suitable for publication and will be formally accepted for publication once it meets all outstanding technical requirements.

Kind regards,

Adam Kane, PhD

Academic Editor

PLOS ONE

Additional Editor Comments (optional):

The authors have addressed all outstanding queries notably the updated tables which have fully fleshed out legends and content.
---

## [Editor Report · Acceptance letter]

2 Dec 2021

PONE-D-21-12399R2 

Why big brains? A comparison of models for both primate and carnivore brain size evolution 

Dear Dr. Chambers:

I'm pleased to inform you that your manuscript has been deemed suitable for publication in PLOS ONE. Congratulations! Your manuscript is now with our production department. 

Kind regards, 

on behalf of

Dr. Adam Kane 

Academic Editor

PLOS ONE